# Mammalian gene expression variability is explained by underlying cell state

Robert Foreman[1,2] & Roy Wollman[1,2,3,*] (iD)

## Abstract

Gene expression variability in mammalian systems plays an important role in physiological and pathophysiological conditions. This variability can come from differential regulation related to cell state (extrinsic) and allele-specific transcriptional bursting (intrinsic). Yet, the relative contribution of these two distinct sources is unknown. Here, we exploit the qualitative difference in the patterns of covariance between these two sources to quantify their relative contributions to expression variance in mammalian cells. Using multiplexed error robust RNA fluorescent *in situ* hybridization (MERFISH), we measured the multivariate gene expression distribution of 150 genes related to $Ca^{2+}$ signaling coupled with the dynamic $Ca^{2+}$ response of live cells to ATP. We show that after controlling for cellular phenotypic states such as size, cell cycle stage, and $Ca^{2+}$ response to ATP, the remaining variability is effectively at the Poisson limit for most genes. These findings demonstrate that the majority of expression variability results from cell state differences and that the contribution of transcriptional bursting is relatively minimal.

**Keywords** $Ca^{2+}$ signaling; gene expression; MERFISH; single cell; transcriptional bursting

**Subject Categories** Chromatin, Transcription & Genomics

**Mol Syst Biol. (2020) 16: e9146**

## Introduction

Gene expression variability is ubiquitous in all biological systems. In multicellular organisms, heterogeneity between different cell types and states confers specialized function giving rise to complexity in whole-system behavior (Raj & van Oudenaarden, 2008; Eldar & Elowitz, 2010; Symmons & Raj, 2016; Suo *et al*, 2018; Tabula Muris Consortium *et al*, 2018). Similarly, single-cell organisms and viruses were shown to utilize heterogeneity at the population level to create diverse phenotypes, such as bet-hedging strategies in changing environments (Veening *et al*, 2008; Vega & Gore, 2014; Rouzine *et al*, 2015). While variability can provide useful functional heterogeneity in a multicellular organism or cell population, it is not necessarily always beneficial (Raj & van Oudenaarden, 2008; Symmons & Raj, 2016). Unregulated stochastic events, i.e., noise, can limit cells' ability to respond accurately to changing environments and can introduce phenotypic variability that can have a negative contribution to overall fitness. Indeed, many biological mechanisms including buffering (Stoeger *et al*, 2016) and feedback loops (Jangi & Sharp, 2014; Schmiedel *et al*, 2015) have been suggested to limit the detrimental effect of gene expression variability. Quantification of the different contributions of mechanisms that cause gene expression variability is an important step toward determining to what degree the variability represents uncontrolled "noise" or cellular stratification and function.

Two key contributors of gene expression variability are allele-specific sources and global factors related to underlying cell state. The analysis of expression covariance between genes is a powerful approach to decompose gene expression variability into these two classes. Landmark works used this approach to investigate expression variability in bacterial cells, which laid a foundation for decomposing variability into allele-specific (intrinsic) sources and variability that originate from sources that affect multiple alleles and relate to the underlying cell state (extrinsic) (Elowitz, 2002; Paulsson, 2005). This work was later extended to yeast (Raser & O'Shea, 2004) and mammalian systems (Raj *et al*, 2006; Sigal *et al*, 2006; Singh *et al*, 2012). The decomposition into allele-specific and cell state components is not always simple. Allele-specific noise in an upstream component can propagate into downstream genes (Sigal *et al*, 2006), whereas temporal fluctuations in the shared components can have nontrivial consequences on expression distributions (Paulsson, 2004; Pedraza & van Oudenaarden, 2005; Shahrezaei *et al*, 2008). Finally, use of the terms "intrinsic" and "extrinsic" is sometimes ill-defined and some models include a "coupled intrinsic" mode as well, which is a form of shared variability and hence "extrinsic" (Rodriguez *et al*, 2019). Despite the sometimes confusing nomenclature, the use of expression covariance to distinguish between allele-specific and shared factors is a powerful decomposition approach.

In addition to covariance-based approaches, the relationship between gene expression distribution variance and mean provides a useful quantitative framework to gain insights into sources of expression variability (Munsky *et al*, 2012). The comparison of

1   Institute for Quantitative and Computational Biosciences, University of California, Los Angeles, Los Angeles, CA, USA
2   Program in Bioinformatics and Systems Biology, University of California, San Diego, San Diego, CA, USA
3   Department of Integrative Biology and Physiology, Department of Chemistry and Biochemistry, University of California, Los Angeles, Los Angeles, CA, USA
    *Corresponding author. Tel: +1 858 210 0905; E-mail: rwollman@ucla.edu

expression variability between genes is not straightforward as expression variance scales with its mean. Three statistical tools are commonly used to describe mean normalized variance: the coefficient of variation (CV), CV squared ($CV^2$), and Fano factor. CV and $CV^2$ are both unitless measures where the CV is defined as the standard deviation divided by the mean and the $CV^2$ is simply the CV squared, or the variance divided by the mean squared. The CV and $CV^2$ are useful to compare the scale of variance between different genes because of their unitless nature. The third measure, the Fano factor, is the variance divided by the mean and therefore not unitless, but it has a special property of being equal to one in the case of a Poisson process. Many biological processes have a variance to mean ratio that is at least Poisson so the Fano factor can define a "standard dispersion", as a result, distributions with Fano factor smaller/bigger than one are considered under/over-dispersed, respectively. Therefore, a simple quantification of the distribution variance scaled by its mean can provide key insights into the underlying mechanism generating the observed distribution (Choubey *et al*, 2015; Hansen *et al*, 2018a).

Multiple studies across bacteria, yeast, and mammalian cells measured over-dispersed gene expression distributions. This observation can have two main interpretations. One interpretation is that the observed over-dispersion is simply a result of the superposition of an allele-specific Poisson variability and cell state variability (Battich *et al*, 2015). The other interpretation is that the allele-specific variability itself is not a simple Poisson process (Suter *et al*, 2011; Dar *et al*, 2015; Corrigan *et al*, 2016; Tantale *et al*, 2016). The latter interpretation was popularized by the introduction of a simple phenomenological model named the two-state or random telegraph model that represented genes as existing in either "on" or "off" states (Peccoud & Ycart, 1995; Kepler & Elston, 2001; Paulsson, 2004; Thattai & van Oudenaarden, 2004; Kaern *et al*, 2005; Friedman *et al*, 2006; Raj *et al*, 2006; Shahrezaei & Swain, 2008; Suter *et al*, 2011; Molina *et al*, 2013; Sanchez & Golding, 2013; Fukaya *et al*, 2016; Lenstra *et al*, 2016). More complex models with multiple states were also considered (Suter *et al*, 2011; Zoller *et al*, 2015; Corrigan *et al*, 2016; Tantale *et al*, 2016; Nicolas *et al*, 2018) but the addition of multiple states does not change the model in a qualitative way. These models suggest that transcription should occur in distinct bursts with multiple transcripts generated when the gene is "on". These two-state models can be described by two overall key parameters: the burst size and frequency that control the resulting gene expression distributions with lower burst frequency and larger burst size contributing to the over-dispersion of the underlying distribution. Overall, both interpretations, bursting and cell state, can explain the observed over-dispersion. There is mounting evidence that for at least many genes, most of the over-dispersion is explained by cell state variables rather than intrinsically noisy transcriptional bursting (Battich *et al*, 2015). Nonetheless, the transcriptional bursting model is still widely used (Larsson *et al*, 2019; Ochiai *et al*, 2019) calling for more systematic investigation.

The relative scales and sources of variability are very important to understand in the modern world of single-cell highly multiplexed measurements. These new technologies are revealing the complex structure of "cell space" with cells occupying a large array of types (Han *et al*, 2018; Rosenberg *et al*, 2018; Tabula Muris Consortium *et al*, 2018), states (Trapnell, 2015; Cheng *et al*, 2019), and fronts (Shoval *et al*, 2012) that reflect functional stratification. Despite our

knowledge that cell types and states manifest as gene expression heterogeneity, sometimes total gene expression variability is interpreted as arising from two-state transcriptional bursting alone (Larsson *et al*, 2019). The gap in our understanding of the relative contribution of cell state and allele-specific factors is hindering progress in assigning functional roles to observed variability (Dueck *et al*, 2016).

To address this knowledge gap, we utilized the two key properties of expression variability: covariance and dispersion. We measured gene covariance and dispersion using joint measurements of individual cells, where for each cell, multiple cell state features were measured, as well as a highly multiplexed measurement of gene expression. We used sequential hybridization smFISH (MERFISH implementation) (Moffitt *et al*, 2016) that allowed us to accurately measure the expression of 150 genes in ~ 5000 single cells. Since expression covariance between genes from the same pathway is higher compared to genes that have distinct functions (Sigal *et al*, 2006; Stewart-Ornstein *et al*, 2012), we focused on a single signaling network and biological function, $Ca^{2+}$ response to ATP in epithelial cells, and a biological response important to wound healing (Funaki *et al*, 2011; Handly *et al*, 2015; Handly & Wollman, 2017). The key advantage of $Ca^{2+}$ response is that the overall signaling response can be measured in < 15 min, a fast timescale that precludes any ATP-induced changes in transcription. Using the combined dataset, we were able to separate the correlated and uncorrelated components using a simple multiple linear regression model guided by the changes in the covariance matrix. We found that after removing all shared components, the remaining allele-specific variability shows very little over-dispersion for most genes measured. Overall, these results indicate that transcriptional bursting is only a minor contributor to the overall observed expression variability.

## Results

To assess the relative contribution of the overall expression variability that stems from allele-specific sources vs underlying cell state variability, we took advantage of the fact that these two sources have different expression covariance signatures. Figure 1 shows simulated data to illustrate how covariance signatures can be utilized to decompose sources of variability. By definition, allele-specific variability is uncorrelated to any other gene, whereas variability that is due to heterogeneity in the underlying cell state will likely be shared between genes with similar function (Fig 1A). When transcriptional bursting dominates (Fig 1B top) the shared regulatory factors will have a small contribution, there will be little correlation between genes and the expression variance will remain largely unchanged after conditioning expression level on any cell state factors (Fig 1B top right). The residual intrinsic variance will have a Fano factor greater than one. On the other hand, when cell state variability dominates (Fig 1B bottom), expression between genes will be highly correlated and conditioning the expression on cell state factors will reduce both the variance and correlation between genes. At the limit, when all shared factors are accounted for, the correlation between genes will approach zero and the Fano factor of the residuals will approach one, the Poisson limit (Fig 1B bottom right). When the contribution of bursting and cell state is

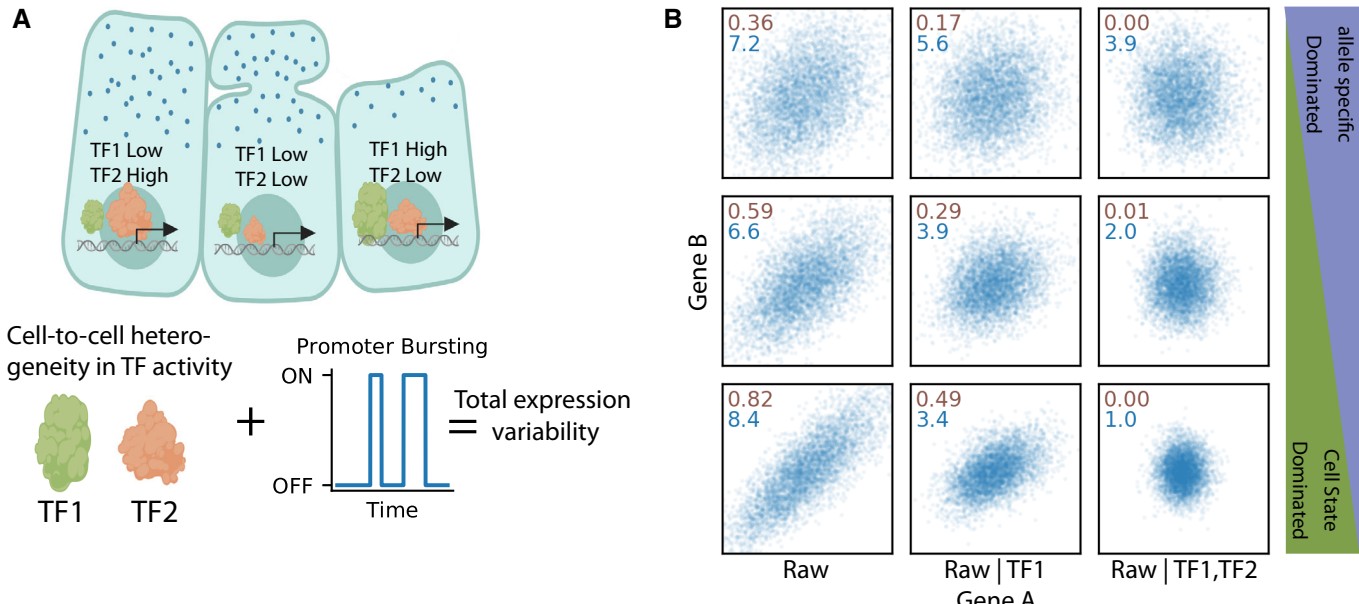

**Figure 1. Transcriptional bursting and trans-acting factors are two distinct causes of cell-to-cell heterogeneity.**

A  Cartoon depicting that different cells can have different activities of trans-factor (TF) regulatory molecules in addition to the effects of transcriptional bursting.

B  Simulated data showing that variability from shared regulatory factors results in correlation between two genes with three example cases: intrinsic dominated noise (top three panels), mixture of cell state and allele-specific sources (middle three), and cell state dominated (bottom three). This correlation is diminished when the expression levels are conditioned on the levels of these shared regulatory factors (middle and right). After conditioning on all trans-acting regulatory factors, the remaining variability due to transcriptional bursting alone is potentially significantly smaller (right). Inset text is the Pearson correlation coefficient between gene A and gene B (brown) and the Fano factor of gene A (blue).

comparable (Fig 1B middle), conditioning on cell state factors will have some effect but the final Fano factor will be higher than one even when the correlation is zero (Fig 1B middle right). Conditioning on cell state factors has a dual effect on correlation and Fano factor, and therefore, it is possible to assess whether the conditioning removed all the obvious extrinsic variability. When all the extrinsic variability is conditioned out, one can confidently interpret whether the residual intrinsic variability is under- or over-dispersed.

To distinguish between the possible situations described above requires accurate highly multiplexed single-cell measurements of gene expression and a sufficient number of cellular features that correlate with the underlying cell state factors controlling gene expression. To achieve this, we developed an experimental protocol that combines MERFISH, a multiplexed and error robust protocol of counting RNA transcripts using fluorescent *in situ* hybridization (Chen *et al*, 2015; Moffitt *et al*, 2016) with rich profiling of the underlying cell state (Fig 2). We used the MCF10A mammary epithelial cell line, which is often used in studies of cellular variability due to their nontransformed nature and their accessibility to imaging (Selimkhanov *et al*, 2014; Qu *et al*, 2015). We focused on genes that share biological function: involvement in the $Ca^{2+}$ signaling network, a key pathway important to the cellular response to tissue wounding (Minns & Trinkaus-Randall, 2016; Justet *et al*, 2019). The two advantages of $Ca^{2+}$ signaling are that (i) we expect that genes that share a function will show a high degree of correlation in their expression levels (Stewart-Ornstein *et al*, 2012). (ii) $Ca^{2+}$ signaling is fast, and we can measure the overall emergent phenotype of the network in < 15 min (Fig 2A), a timescale faster

than that of gene expression in mammalian cells (Shamir *et al*, 2016). In our protocol, cells were rapidly fixed after live cell imaging (10–15 min from ATP stimulation to fixation, Appendix Fig S1), and therefore, the gene expression measured in the same cell is unlikely to have changed as a result of the agonist.

MERFISH is a multiplexing scheme of smFISH where transcript identity is barcode-based, and the barcodes are imaged over several rounds of hybridization. During each hybridization round, dye-labeled oligos are hybridized to a subset of RNA species being measured, the sample is imaged, and RNA appears as diffraction-limited spots; then, the dye molecules are quenched, and the process is repeated until all barcode "bits" are imaged. By linking diffraction-limited spots across imaging rounds, we can decode the RNA barcodes by identifying the subset of images where a bright diffraction-limited spot appears at the same *XYZ* coordinate (Fig 2B). The use of combinatorial labeling allows exponential scaling of the number of gene images with the number of imaging rounds. The scaling is mostly limited by the built-in error correction (Chen *et al*, 2015). In this experiment, we used 24 imaging rounds (eight hybs × three colors) where each RNA molecule was labeled in four imaging rounds. An example of the MERFISH data is shown in Fig 2B. Overall, we measured the expression of 150 genes including 131 genes annotated as involved in $Ca^{2+}$ signaling network (Kanehisa & Goto, 2000; Bandara *et al*, 2013; Kanehisa *et al*, 2019), 17 genes to mark stages of the cell cycle (Whitfield *et al*, 2002), and two genes that correlate with the sub-differentiated state of MCF10A cells (Qu *et al*, 2015). We estimate our detection efficiency to be ~ 95.5% and false-positive rate < 1% per gene per cell. Overall

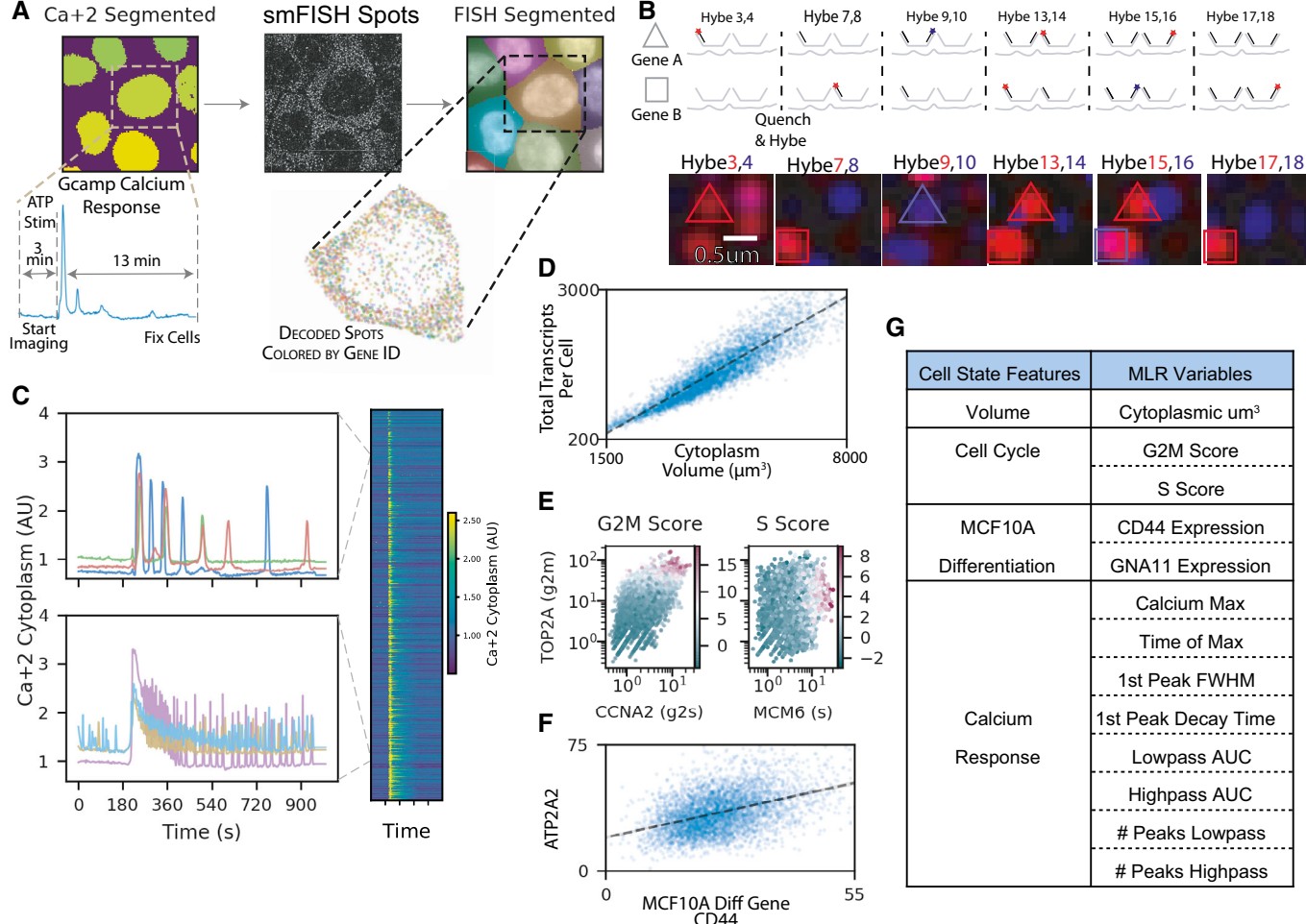

**Figure 2. Paired single-cell MERFISH and live cell calcium imaging.**

A   Experimental overview—live cells are imaged for their calcium response to ATP before being fixed and imaged to measure gene expression of 150 genes.

B   smFISH spots are imaged over several rounds of hybridization and aligned such that individual genes are encoded as specific series of dark and bright spots throughout all rounds of hybridization.

C   Left, representative calcium trajectories demonstrating the heterogeneous response to ATP stimulation, top vs bottom left. The right panel is an image plot of all 5000+ successfully paired to smFISH cells.

D   Cellular volume is measured and the correlation between total transcripts per cell and the cellular volume is shown.

E   Left, shows marker gene expression for cell cycle-related genes used to derive a g2m score (coloring). Right, is the same as the left panel with a representative gene used to derive the S score for each cell.

F   Correlation of a representative gene (ATP2A2) with a gene that marks the differentiation status of MCF10A cells (CD44).

G   Table of the cell state features categories and the complete list of the 13 factors used in the multiple linear regression (MLR) statistical model.

spearman correlation with bulk RNAseq was 0.84 (Appendix Fig S3).

Our decomposition into allele-specific and cell state components is based on conditioning on multiple cell state factors. While it would be ideal to directly measure the regulatory factors that causatively control gene expression variability, more accessible measurements, e.g., cell size or cell cycle stage, that are correlated with these causative regulatory factors are sufficient for the conditioning process. Given that the genes we probe are related to $Ca^{2+}$ signaling, we first extracted key features from time series of cytoplasmic $Ca^{2+}$ response measured with a calibrated GCaMP5 biosensor (Appendix Fig S2A). The live cell imaging of cytoplasmic $Ca^{2+}$ levels (Fig 2C) showed a highly heterogeneous response,

qualitatively and quantitatively similar to previous work on $Ca^{2+}$ signaling in MCF10A cells where we observed a mixed population response with a wide range of response phenotypes (Yao *et al*, 2016; Handly & Wollman, 2017). We used a feature-based representation of $Ca^{2+}$ response to represent cellular factors that we anticipate correlate with underlying cell state (Fig 2G and Appendix Fig S2). In addition to $Ca^{2+}$ features that are specific to $Ca^{2+}$ signaling, we also measured a few global features of the cell that are likely to be correlated with expression changes of most genes. Specifically, we measured cell volume, cell cycle stage, and two markers of MCF10A differentiation status (Fig 2D–F). As was shown in the past, cell volume strongly correlated with the total number of transcripts per cell (Fig 2D) indicating that at least for some genes, cell

state factors must be important contributors to their expression variability (das Neves *et al*, 2010; Shalek *et al*, 2014; Battich *et al*, 2015; Padovan-Merhar *et al*, 2015; Hansen *et al*, 2018a). However, not all genes show the same strength correlation with volume, and some cell cycle genes are more complexly related to volume (Appendix Fig S4). Similarly, the cell cycle stage and MCF10A differentiation status were correlated with specific genes (Fig 2E and F) (Buettner *et al*, 2015). Overall, we measured 13 different cellular features that will be used to decompose variance in all 131 $Ca^{2+}$-related genes we measured. By focusing on a smaller number of specific features that relate to the $Ca^{2+}$ response augmented by established global cell state features like cell size and cell cycle state, we expected to be able to capture most of the expression variability that comes from underlying cell state heterogeneity. These results are consistent with previous work demonstrating widespread cell cycle and differentiation-related variability in the transcriptome (Battich *et al*, 2015).

To decompose the observed expression into multiple components, we used standard multiple linear regression (MLR) (Battich *et al*, 2015; Hansen *et al*, 2018a). Figure 3A shows the scatter plots of expression of two representative genes (ATP2A2 and RRM1) plotted against cell volume, cell cycle, differentiation markers, and $Ca^{2+}$ feature. The scatter plots show that (i) there is indeed a correlation between expression and some of these cell state features. (ii) The

amount of variance that is explained by each cell state feature can change between genes. Overall, the simple MLR model with 13 independent measurements was able to explain between ~ 15 and 85 % of the observed variance with a median of 0.62 (Fig 3B). To assess the relative contribution of each cell state feature, we looked into the relative fraction of explanatory power for each feature category (Fig 3C). Overall, cell volume has the most explanatory power, but for some genes, cell cycle and $Ca^{2+}$ features contribute meaningfully to the explained variance. While some of the features had a small effect in terms of the overall variance explained by the feature, in most cases, the effects were very unlikely to be a result of pure random sampling, permutation-based statistical testing showed that most genes measured here are statistically correlated with at least one calcium feature (Fig 3D).

A key uniqueness of our approach is that gene expression is measured in a multiplexed fashion allowing the estimation of the correlation between genes. Figure 4A shows the correlation matrix of the raw counts, and the counts conditioned on cell state features. As expected, as we increase the number of cell state features included in the MLR, the overall gene-to-gene correlation goes down. Interestingly, the full MLR model that only includes 13 identical terms for all genes is able to reduce the overall correlation between genes significantly. To quantify the bulk correlation, we measured the amount of variance that is explained by the first two

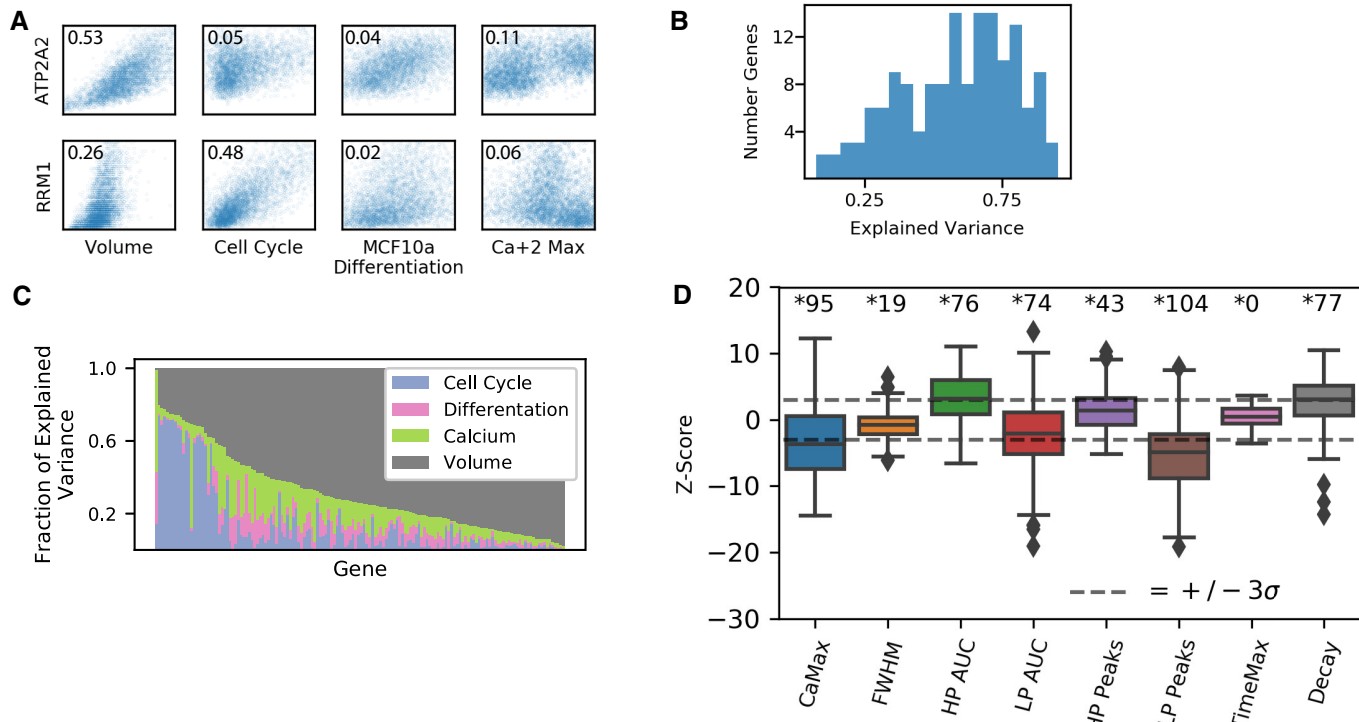

**Figure 3. Decomposition of gene expression variability using multiple linear regression.**

A   Representative scatter plots of correlation between two individual genes (rows) and different cell state factors (columns). The percent of variance explained by each factor in the MLR model for each gene is annotated in the corner.
B   A histogram of the overall explained variance for each gene.
C   Stacked bar plot showing which cell state feature categories contribute to the explained variance of the MLR.
D   The significance of calcium features for 150 genes was estimated by *Z*-scoring the slope of the feature in a null distribution of 1,600 bootstrapped shuffled data slopes. The number of statistically significant genes for each feature is shown above [adjusted *P*-value (Bonferroni) < 0.05]. Whiskers are at 1.5 times the interquartile range.

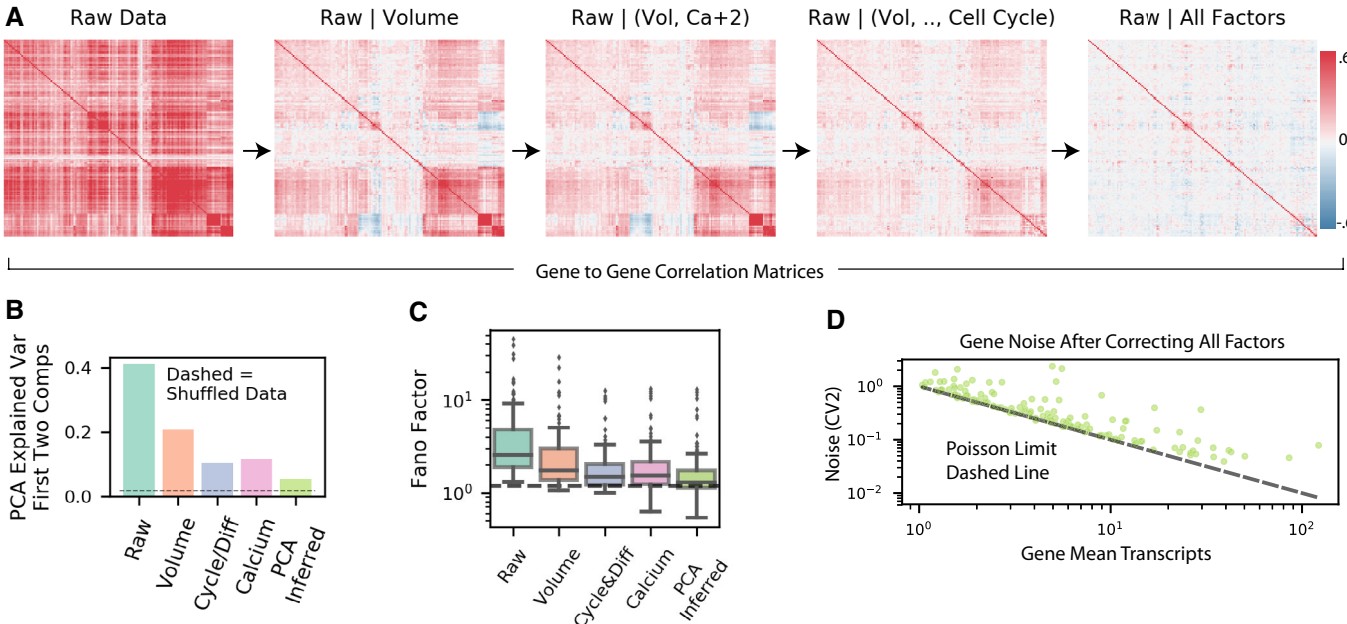

**Figure 4. Residual variability from MLR models contains significantly less covariation between genes and close to Poisson variability within individual genes.**

A   Gene–gene correlation matrices showing the reduction of covariance after conditioning on cell state features.

B   Explained variance of first two components of PCA for each stage of MLR models showing reduction in shared variability with increasing number of cell state factors. The dotted line shows the variance explained by first two PCA components when the data are shuffled.

C   Fano factor distributions of 150 genes measured at different levels of cell state conditioning are shown as boxplots. The whiskers are at 1.5 times the interquartile range. Dashed line is the Poisson expectation with technical noise.

D   Scatter plot of residual gene expression coefficient variation squared for each gene after decomposition of all cell state features. Poisson expectation is shown as dashed line.

components of a principal component analysis (PCA; Fig 4B). Without conditioning on any cellular feature, the first two components explain > 40% of the variance. This is reduced substantially to < 10% of the overall variance, in the full MLR. The substantial reduction in the gene-to-gene correlation demonstrates that we were able to condition away most of the shared components. Still, the remaining correlation was not completely removed, and therefore, we added another term to the model that is based on the first two principal components of a PCA after taking all other features into account. These two components most likely represent some cell state features that were not sufficiently captured by our 13 cellular features. With the addition of the last "hidden" feature, the overall variance that is shared is very close to values from shuffled data. Overall, the analysis of expression covariance demonstrates that our simple MLR sufficiently captures most of the information related to cell state that is required for conditioning expression distribution.

Finally, we wanted to determine the overall dispersion remaining in the allele-specific gene expression distribution. The allele-specific variability is estimated as the residual variability in the raw gene expression counts after conditioning on cell state factors. As we increase the number of cell state features we conditioned on, we saw a substantial reduction in the distributions of Fano factor magnitudes (Fig 4C). When all 13 cell state features and the two hidden features estimated based on PCA are included, the Fano factor is very close to one for most of the genes. Note that we do not perform any correction for technical noise; so, the limit of one is only theoretical. Similarly, analysis of the $CV^2$ vs the expression means on a log–log plot shows that all genes are very close to the

Poisson limit (Fig 4D). The proximity to the Poisson limit is similar across all expression levels. Therefore, these data indicate that super-Poissonian transcriptional bursting plays a very minor role in allele-specific variability. It is unclear whether the few genes that do show over-dispersion whether they have significant levels of transcriptional bursting or whether our conditioning procedure failed to sufficiently remove cell state effect.

## Discussion

Here, we analyzed the relative contribution of gene-specific variability that arises from transcriptional bursting, i.e., episodic synthesis of multiple transcripts from a gene, and variability that is shared among multiple genes. Our approach is enabled by very rich single-cell measurement that include live cell $Ca^{2+}$ response to ATP, global cell state factors such as size and cell cycle stage, and the expression level of 150 genes all in the same single cells. Using these data, we were able to decompose gene expression variability into gene-specific and cell state components. We show that after removing covariability from gene expression distributions, the remaining variability follows a simple Poisson model. The residual allele-specific variability is not over-dispersed and therefore not consistent with models of transcriptional bursting where a gene is actively transcribed only during a small fraction of time.

The popularity of the transcriptional bursting model is evident by the large number of papers that fit the entire RNA and protein distributions to the two-state model without considering other sources of

variability (Skupsky *et al*, 2010; Suter *et al*, 2011; Molina *et al*, 2013; Dey *et al*, 2015). In other cases, cell state was considered using dual reporters (Sigal *et al*, 2006; Strebinger *et al*, 2018), assuming timescale separation (Dar *et al*, 2012), or conditioning on forward scatter (Sherman *et al*, 2015). However, without multiplexed expression measurements it is difficult to determine whether conditioning on cell state was done to completion. The high goodness of fit of the two-state model to uncorrected or partially corrected distributions that shows substantial bursting could simply be a case of over-interpretation of model fit. RNA-binding systems, such as MS2, allow direct live cell observation of transcription bursting, and many groups have observed burst-like punctuated transcription (Muramoto *et al*, 2010; Ferguson & Larson, 2013; Corrigan *et al*, 2016; Fritzsch *et al*, 2018). While direct visualization is compelling, it is unclear whether punctuated transcriptional events are due to stochastic transition of promoter state, as suggested by two-state model, or due to stochasticity in the activity of an upstream regulatory element. Furthermore, difficulty in quantifying the number of mRNAs synthesized in each such event makes it difficult to distinguish between a two-state model and a one-state model with a low rate of transcription that will generate a Poisson distribution. In fact, our results are consistent with recent measurements that showed that TTF1 mRNA is generated in "bursts" of 1–2 mRNA (Rodriguez *et al*, 2019). Furthermore, the two alleles of TTF1 showed coordination between these bursts suggesting that the observed transcriptional events are coupled through trans-regulatory factors. Finally, temporal changes in global rates of transcriptions (Skinner *et al*, 2016; Shah *et al*, 2018) can also make the interpretation of a single allele temporal reporter challenging. It is important to note that our work focuses on genes that encode for calcium signaling activity and might not represent all genes, such as reporters controlled by viral promoters (Singh *et al*, 2010; Dar *et al*, 2012) and genes that are key to cellular differentiation (Hansen & van Oudenaarden, 2013; Ochiai *et al*, 2014). Overall, it is advisable to use more caution when interpreting gene expression variability as evidence of transcriptional bursting.

Our measurements are based on cytoplasmic RNA, and it is possible that mechanisms related to RNA processing reduce the dispersion of RNA distribution in the cytoplasm after it was generated in an over-dispersed manner through bursting (Battich *et al*, 2015). Cells include a large number of RNA-binding proteins many with unknown function, and it is possible that some function as part of post-transcriptional noise reduction mechanisms (Hansen *et al*, 2018b). However, some of the proposed mechanisms such as nuclear export of RNA were shown to act as amplifiers of observed dispersion (Hansen *et al*, 2018a). For different genes, there can be different effects explaining why observed cytoplasmic transcript counts are distributed approximately Poisson for most genes, despite widespread observation of bursts during transcription. Expression variability could be buffered by processes such as nuclear export (Stoeger *et al*, 2016; Chen and van Steensel 2017; Xia *et al* 2019), bursting may not occur for all genes (Berry *et al* 2017), and bursting may be linked to extrinsic fluctuations in enhancer activity rather than intrinsic noise (Fukaya *et al*, 2016). Therefore, the degree by which post-transcriptional mechanism can be used to reduced expression noise is an important open question. Until additional data will help clarify the ubiquity of such mechanisms, the most parsimonious interpretation is simply that RNA synthesis does not happen in large allele-specific bursts.

Recent technological advances in the ability to measure single-cell gene expression with scRNAseq and sequential smFISH approaches are providing an unparalleled view into the underlying "cell state space". The distribution of cells in "cell state space" and the definition of cell types and states within this space are key open research areas that will likely to further grow in importance with further improvements in single-cell measurement technologies (Wagner *et al*, 2016; Eng *et al*, 2019). Our work has two important implications on our understanding of this "cell state space", at least with regard to the heterogeneity of a single cell type: (i) All the shared variability was reduced using only a simple representation of cell state as 13 linear coefficients. Furthermore, most of these 13 features had only a very small contribution to the overall explanatory power suggesting that cell state distribution can be represented by few latent dimensions. An observation that emboldens efforts to learn the cell state manifolds (Moon *et al*, 2018). (ii) Expression noise, i.e., unregulated variability in gene expression that is a result of stochastic biochemical interactions in effect defines a "resolution limit" of the cell state space. Our results indicate that the highly heterogeneous distribution of cells within cell state space is likely not due to the inability of cells to control their expression levels rather our work indicates functional stratification of cells within this space. Collectively, these contributions pave the way to a more rigorous definition of cell state that is based on concepts of signal to noise where the signal is represented by regulated differences between cells and noise is due to unregulated stochastic events. Such definitions will help identify the functional role of cellular heterogeneity.

# Methods

### Contact for reagent and resource sharing

Further information and requests for resources and reagents should be directed to and will be fulfilled by the Lead Contact, Roy Wollman (rwollman@ucla.edu).

### Experimental model and subject details

The MCF10a cells used in this study are Homo Sapien, female cells with the RRID: CRL-10317. This cell line has not been authenticated, but bought directly from ATCC. Cells were grown in complete media: DMEM/F12 media (Gibco) supplemented with 5% Horse Serum (Life Technologies), EGF 20 ng/ml, hydrocortisone 0.5 μg/ml, cholera toxin 0.1 μg/ml, insulin 10 μg/ml, and Penicillin/Step 100 U/ml referred to as complete media.

### Cell culture

MCF10a cells were grown in complete media (above) and passaged at 70–90% confluency. Cells were seeded onto coated 40 mm #1.5 coverslips (Bioptechs) and grown to confluence in 5-mm-diameter PDMS wells before changing media to complete media without EGF and 1% horse serum, instead of normal 5%, 6–8 h before imaging. Coating solution consists of sterile-filtered 10 μg/ml fibronectin, 10 μg/ml bovine serum albumin, and 30 μg/ml type I collagen in DMEM.

## mCherry GCamp5 fusion construct creation

For pPB-mCherry vector construction, a PCR product encoding GCaMP5 sensor incorporating the CaMP3 mutation T302L R303P D380Y and no stop codon (Addgene plasmid #31788) was directionally ligated into pENTR/D-TOPO vector (Invitrogen K243520) resulting in pEntry_GCaMP5G construct.

(For: caccATGGGTTCTCATCATCATCATCATCATGGTATGGCTA GCATGAC, REV: TTACTTCGCTGTCATCATTTGTACAAACTCTTCG TAG) pEntry_GCaMP5G was linearized with PCR using standard Phusion® Hot Start Flex 2X Master Mix (NEB Cat# M0536L) protocol (FOR: cgcgccgacccag, REV: ctcgagggatccggatcctcccttcgctgt catcatttgtacaaac). PCR product was then subjected to DpnI digestion (NEB cat# R0176S) and gel purification with Zymoclean Gel DNA Recovery Kit (ZYMO cat#D4001). A sequence encoding mCherry and a5′ linker was PCR-amplified (FOR: gaggatccggatccc tcgagAccatggtgagcaagggc REV: aagaaagctgggtcggcgcgcttgtacagctcgt ccatg). mCherry2-C1 was a gift from Michael Davidson (Addgene plasmid # 54563).

GeneArt Seamless Cloning and Assembly Enzyme Mix (Invitrogen cat# A14606) was used to assemble a construct encoding for GCaMP5 sensor fused with a short linker to mCherry called pENTRY-GCaMP5fusedmCherry. LR recombination between this entry clone and a custom gateway PiggyBack transposon vector with 1 µl LR Clonase II enzyme (Invitrogen: cat #11791020) resulted in the final construct of pPB_CAG_GCaMP5-fusedmCherry_blast.

## mCherry GCamp5 fusion MCF10A cell line creation

To generate stable cell lines constitutively expressing cGamp5fusion-mcherry, MCF10A cells were grown in the standard conditions and co-transfected using Neon transfection system (Invitrogen cat#MPK1025) and transposase expression vector pCMV-hyPBase (Sanger Institute) in the 4:1 ratio with 0.625 µg of transposase and 2 µg of transposon plasmid per well in six-well dish. Electroporation parameters were as follows:

Pulse voltage (v): 1,100
Pulse width (ms): 20
Pulse number: 2
Cell density (cells/ml): $2 \times 10^5$
Transfection efficiency: 45%
Viability: 65%
Tip type: 10 µl

Stable, polyclonal cell populations were established after blasticidin selection (10 µg/ml).

## Coverslip modification

Forty millimeter coverslips (Bioptechs) were allyl silane functionalized according to Moffitt et al (2016), which briefly consists of washing coverslips in 50% methanol and 50% 12M HCl, and then incubating at room temperature in 0.1% (vol/vol) triethylamine (Millipore) and 0.2% (vol/vol) allyltrichlorosilane (Sigma) in chloroform for 30 min. Wash with chloroform and then with 100% ethanol, and air-dry with nitrogen gas. These were stored in a desiccator for less than a month until use.

## Calcium imaging

Cells were stained with 0.1 µg/ml Hoechst for 20 min and then rinsed with imaging media. Each well was imaged and stimulated consecutively as follows: image 3 min of Gcamp before stimulating with 6 µM ATP in imaging media and then imaged for another 13 min. Gcamp was imaged every 2–3 s, and Hoechst was imaged every 4 min for segmentation. Immediately following imaging of a well, that well was fixed with 4% formaldehyde in PBS. The next well was imaged, and then, the previously imaged/fixed well was washed 3× with PBS.

## Sequential FISH staining

PDMS wells were removed, and cells were briefly fixed for 2 min, washed 3× with PBS, and then permeabilized with 0.5% Triton X-100 in PBS for 15 min. Coverslips were washed 3× with 50 mM Tris and 300 mM NaCl (TBS), and then immersed in 30% formamide in TBS (MW) for 5 min to equilibrate; all the liquid was aspirated from the petri dishes; and 30 µl of 75 µM encoding probes and 1 µM locked poly-T oligos were added on top of the coverslip, and a piece of parafilm was place on top of the coverslip to evenly spread the small volume over the surface and prevent evaporation. The entire petri dish was also sealed with parafilm and incubated at 37°C for 36–48 h. The parafilm was removed, and the coverslip was washed 2× with MW buffer with 30-min incubation at 47°C for both washes. A 4% polyacrylamide hydrogel was then cast to embed the cells before clearing with 2% SDS, 0.5% Triton X-100, and 8 U/ml proteinase k (NEB P8107S), according to previously published methods. Coverslips were incubated in clearing buffer for 24 h and then washed 3× in TBS for 15 min each at room temperature (Moffitt et al, 2016).

## Sequential FISH imaging

smFISH staining was imaged on a custom-modified Zeiss Axio Observer Z1 body with Andor Zyla 4.2 sCMOS camera and 1.4NA 63 Plan-Apo oil immersion objective. Illumination light was provided by LUXEON rebel LEDs (deep red, lime, blue, and royal blue) to excite Cy5, Atto 565, Alexa 488, Hoechst, and 200 nm deep blue fiducial markers. The microscope was controlled by micromanager (Ausubel et al, 2001) and custom MATLAB software. Automated washing during sequential rounds of hybridization was accomplished by using a previous published setup (Moffitt et al, 2016; Moffitt & Zhuang, 2016). Briefly, FCS2 bioptech flow chambers were attached to a Gilson Minipuls peristaltic pump pulling liquid from reservoirs attached to Hamilton MVP valves. The pump and valves were controlled with Arduino, and serial commands with Python https://github.com/ZhuangLab/storm-control/tree/master/storm_control/fluidics. This setup was used to automatically wash cells with TBS, then 2 ml of TCEP (Sigma) in TBS incubated for 15 min, then rinse with TBS, then flow in 2 ml of wash buffer [10% ethylene carbonate in TBS with 2 mM vanadyl ribonucleoside complex (NEB)], followed by 3 ml 3 nM readout probes in wash buffer incubated for 15 min, then rinsed with 2 ml wash buffer, then 1 ml of TBS, and finally 3 ml of imaging buffer. Imaging buffer is 0.15 U/ml rPCO (OYCO), 2 mM PCA (Sigma), 2 mM Trolox (Sigma), 50 mM pH 8.0 Tris–HCl, 300 mM NaCl, and 40 U/ml murine RNase inhibitor (NEB).

## FISH oligo pool design amplification

Oligopools were ordered from CustomArray. The oligos were designed using previously published software (Moffitt & Zhuang, 2016). Briefly, design involves selecting 30-bp regions with 40–60% GC for each target gene that maximizes specificity of the oligo by finding shared 15-mer substrings against all other transcripts in the human genome. These regions are concatenated with sequences for three readout probe binding sequences and flanking 20 bp primers.

Probes were amplified according to another previously published work (Wang *et al*, 2018). Briefly, we performed limited cycle qPCR with a T7 promoter on the reverse primer. The PCR was terminated one cycle after saturation during the extension phase. PCR product was column-purified, and then, *in vitro* transcription further amplified the oligos (NEB Quick High Yield Kit); t7 reactions were purified with desalting columns, and converted to ssDNA with Maxima RT H-(Thermo).

## Gcamp image processing

Cell nuclei were segmented using custom Python 3.6 scripts. Cell nuclei were segmented using the Hoechst staining. Nuclear images were low-pass filters with Gaussian of sigma 5 pixels. Then, regional maxima were found with corner_peaks from scikit-image; these peaks were used as seeds in a watershed of the negative intensity of the images, and thresholded with Otsu of the smoothed nuclear images. This was repeated for each time point, and the centroid of each nuclear mask was tracked across time using linear assignment. Segmented nuclei were used as masks to calculate the mean intensity within each cell mask in the Gcamp channel and also the channel for mCherry-fusion expression marker for Gcamp. Finally, Gcamp values were divided by the mCherry values to give expression normalized calcium trajectories.

## Calcium trajectory feature extraction

Calcium trajectories were processed with wavelets to find low-pass, smoothed, and high-pass trajectories by thresholding coefficients of different scale wavelets. Peaks were detected in the low pass and high pass with scipy's find_peaks and prominence thresholds of 0.1 and 0.15, respectively. Decay time of the first major peak after ATP stim was calculated; FWHM of the first peak after ATP was calculated; the AUC of high pass and low pass was calculated with numpy's trapz; the maximum of each calcium was calculated; and the time of maximum was also calculated from smooth trajectories.

## Alignment to live cell images

EM microgrids (G400F1-Cu EMS) were glued (23005 Biotium) to 40 mm Bioptech coverslips. These grids were imaged in brightfield to determine the stage coordinate of fiduciary marks on the microgrids. A rotation and translation transformation was fitted between the live cell and smFISH coordinates of microgrid fiduciary marks. This ensured that we imaged the same FOVs, but additional alignment was performed after imaging. smFISH images were downscaled until they had a pixel size matching the live cell imaging (63× vs 10× with same Andor Zyla Camera so 6.3× downscaling). Cross-correlation template matching with live cell templates and smFISH

candidate images was performed iteratively with range of rotational angles (−5 to +5 degrees) in order a second set of "image" translations and rotations that maximize the cross-correlation scores. A threshold was then applied, and downsampled images were stitched together and overlaid to confirm successful alignment.

## smFISH image alignment

All rounds of hybridization contained 200 nm blue beads (F8805 Thermo Fisher) that were imaged in addition to smFISH oligos.

First, the coordinates of putative beads were determined with subpixel accuracy by upsampling images by a factor of 5 (~ 20.5 nm pixel size) and finding peak coordinates of normalized cross-correlation between a Gaussian "bead template" and bead images in 3D. Next, a translational transformation was estimated from these putative beads with a custom algorithm designed to be robust to false detection of beads. Briefly, neighborhoods of beads with a radius of maximum shift (100 pixels) were found and the differences each of these pairs were calculated. Next, the bead coordinate differences were density clustered and bead pairs from the largest cluster were used in a least sq error optimization of translation vector that minimizes residual of all bead pairs after translation. This fit was performed in 3D, and any FOVs with a residual > 0.5 pixels *XY* or 1.2 μm (3 frames) in *Z* were discarded.

## Chromatic aberration correction

TetraSpeck (4-color) 100 nm beads were imaged in all channels used for smFISH imaging. The subpixel centers of these beads were found as described above, and the misalignment of channels was calculated as a function of the *XY* image coordinate. Images were then interpolated in 2D to correct for systematic differences between channels. (Mostly only necessary at edges of the images due to large camera sensor size.)

## Gene calling

Spots were called with a reimplemented algorithm deeply inspired by Jeffrey R. Moffitt *et al* (2016), and code is available at https://github.com/wollmanlab/PySpots. Images were taken every 0.4 μm in *Z*, but groups of three images one above and below the current *Z* slice being processed were maximum projected to form a pseudo-*Z* slice to be further processed. Then, two *Z* slices were skipped before forming another pseudo-*Z* slice. These local max projections help gene calling perhaps do to making the imaging more robust to misaligned images, or uncorrected planarity issues in the objective. Second, fiduciary 200 nm beads were used to fit *XYZ* translation transformations described in the image alignment section, and all pseudo-*Z* slices were warped to correct for chromatic aberration and translations from stage reproducibility error. Registered and chromatic aberration fixed images were then high-pass-filtered by subtracting a Gaussian convolution with sigma 2.2 pixels from the original images. These high-pass-filtered images were then deconvolved for 20 iterations of Lucy–Richardson deconvolution using the flowdec package (Czech *et al*, 2018). Finally after deconvolution, the images were blurred by Gaussian convolution with a sigma of 0.9 pixels. The output at this step for each site imaged is a matrix of (2,048, 2,048, 24, #*Z*) elements. Where 2,048 is the image width and height, 24 is the number of codebits

used to encode gene identity (three colors × eight rounds sequential hybridization) and #Z is the number of pseudo-Z slices. Next, each Z slice was processed separately on a per pixel basis to assign each pixel as its gene identity or as background. This process was done by dividing each of the 24 images by the 95th percentile of that image to make the intensities for different codebits more similar, and L-2 normalizing each pixel. Then for each pixel, the Euclidean distance to L-2 normalized codebit vectors was calculated, and if that distance was less than the volume of a nonoverlapping hypersphere for all codewords (0.5176), then the pixel was classified as that closest codeword. This approach is essentially testing whether the intensities from all 24 codebits point in the direction of a particular codeword in 24-dimensional space. Finally, these classified images (2,048, 2,048, #Z) were segmented to collect groups of connected components with that same gene label. Finally, genes calls were thresholded on the number of pixels for each group of connected components and the average intensity of the set of connected components.

## Calculation of cell volume

A 3-D histogram of gene calls for each cell was calculated and smoothed with a Gaussian filter of 10 pixels. The number of voxels (1, 1, 1 μm) with at least 0.5 RNA was calculated and used as the volume for each cell.

## Estimation of MERFISH gene calling error rates

### False-negative rate (sensitivity)

For each XY centroid of spots that were called genes, the intensity vector was extracted from the 24-bit codestack. The Euclidean distance between these intensity vectors was calculated for five items: the actual codeword (24-bit binary vector with exactly four one bits) and each of the four possible 1-bit dropouts (24 for binary vectors with exactly 3 one bits). These 1-bit dropouts comprise synthetic codewords that are still uniquely close to the true gene than any other gene in the codebook. For each gene, spot we enumerate whether the actual intensity vector was closer in Euclidean distance to the full hamming weight 4 codeword, or one of the 1-bit (hamming weight 3) dropout codewords for the same gene. From this, we calculated the frequency of gene calls that were closest to the hamming weight 4 codeword, and normalizing by the total number of gene calls, we get 0.68. That is, $1–0.68 = 32\%$ of the gene calls are expected to contain a 1-bit dropout. If we consider 0.68 the probability that we detect all four bits of a gene, and given that we can successfully classify a gene that we only detect three bits: overall detection sensitivity is $p^4 + 4*(1 − p)*p^3 = 0.955$ where $p^4 = 0.68$ and $p = 0.68^{1/4} = 0.908 = P$ (detect a bit of codeword).

### False-positive rate

The encoding capacity of our codebook is 472 genes/codewords, but we only assign 336 of them to actual genes in this experiment. The other 144 codewords are reserved as "blank codewords". We allow these blank codewords to be classified by the algorithm as if they were real genes. Then using the counts of these blank codewords, we estimate the false-positive rate of our classification algorithm as (# blank codeword counts)/(144 possible barcodes) = rate of false positives per barcode. We normalize the rate per barcode to the number of cells segmented to get an expected average false-positive rate of < 1% per gene/cell.

## Simulation of gene variance decomposition

For each of the three combinations of cell state and allele-specific noise simulations, there were three transcription factors and two genes simulated. Transcription factors were Poisson distributed, and genes were simulated as gamma distributions with shapes dependent on additive combinations of transcriptions 1, 2, and 3. The scale of the gamma distributions was varied to control the amount of "allele-specific variability", and the amount of gene correlation was controlled by the fraction of shape shared between genes.

For each of the three combinations of different noises, there were four linear models fitted using python statsmodels ols package. For each gene, a model was fitted for gene ~ tf1 and gene ~ tf1 + tf2. Then, the residuals from the fit were adjusted by adding back the mean of expression for that gene, and these mean adjusted residuals are the distribution of the gene conditioned on tf1 or (tf1, tf2).

## Cell cycle features

Cell cycle features were calculated using the scanpy package (Wolf *et al*, 2018).

## Gene variance decomposition

The same method (linear model residuals) as in the simulation was used to decompose variance for gene expression. In order to investigate residual correlations between genes with different sets of conditioning variables, the decomposition was repeated from different combinations of feature combinations. The first stage involved only gene ~ volume, and then gene ~ volume + s_phase + g2 m_phase… finally for the inferred features, we used PCA components #1 and #2 as features: gene ~ pca_comp1 + pca_comp2.

## Statistical test of calcium feature significance

Volume-adjusted gene expression counts were fitted with a linear model based on calcium features. For every gene separate and every calcium feature separately, a shuffled linear model was also calculated. That is, for each calcium feature and gene many bootstrap models were estimated where a single calcium feature was shuffled and the model was fitted. The slopes of these fitted models on shuffled data formed a null distribution, and then, the P-value of the feature for that gene was considered [100-Qtile(unshuffled slope in shuffled bootstraps)] where 0 is 1/#Bootstraps.

# Data availability

All raw data are available on figshare at https://doi.org/10.6084/m9.figshare.11410212.v1.

**Expanded View** for this article is available online.

## Acknowledgements

We are thankful to Jeffrey Moffitt for his extensive feedback about the MERFISH method and Anna Pilko for making the mCherry-Gcamp construct and cell line. This work was funded by NIH grants to RW EY024960 and GM111404.

## Author contributions

RF and RW conceptualized the experiments and data analysis. RF performed the experiments and performed data analysis. RF and RW wrote and edited the paper.

## Conflict of interests

The authors declare that they have no conflict of interest.

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
