## [Review Process File · Molecular Systems Biology]

Mammalian gene expression variability is explained by underlying cell state

Robert Foreman and Roy Wollman.

Review timeline:

Submission date:	31 st July 2019
Editorial Decision:	27 th September 2019
Revision received:	1 st November 2019
Editorial Decision:	11 th November 2019
Revision received:	4 th December 2019
Accepted:	7 th January 2020

Editor: Maria Polychronidou

Transaction Report:

1st Editorial Decision

27th September 2019

Thank you again for submitting your work to Molecular Systems Biology. We have now heard back from the three referees who agreed to evaluate your study. As you will see below, the reviewers acknowledge that the study is timely and the conclusions seem interesting. They raise however a series of relatively minor concerns, which we would ask you to address in a revision.

Most of the reviewers' comments refer to the need to perform text modifications and to provide further clarifications. A rather important concern refers to the need to better contextualize the study in comparison to previous work. Please feel free to contact me in case you would like to discuss in further detail any of the issues raised by the reviewers.

REFERE REPORTS

Reviewer #1:

Review of 'Mammalian gene expression variability is explained by underlying cell state' by Foreman and Wollman.

This is an exciting paper that employs multiplexed single-molecule FISH to study sources of variability in mRNA abundance. I find the work of very high quality, and the results interpreted justly and of great significance to the quantitative single-cell biology community, and thus highly appropriate for a journal such as Molecular Systems Biology. I therefore recommend it for publication, and request only revisions related to the text, as I explain below.

In general, I find that the authors can do a better job in placing their findings within the context of previous work. Given the far-reaching implications that their findings, and similar findings of

previous studies, have on the gene expression noise field, and on the single-cell biology field in general, it is important that these findings are repeatedly communicated and in the most visible places. This does not diminish their work, on the contrary, it makes it stronger, and particularly for a journal like *Molecular Systems Biology*, I would expect this.

On page 3, the authors state that: 'Overall both interpretations, bursting and cell state, can explain the observed over-dispersion and it is currently unclear which one is correct.' I understand why the authors pose the problem like this, as it is a nice lead into their ambition to provide insight into which interpretation is correct. But in essence, the former point of view has been systematically revised by many studies into one that more fits the latter point of view (exceptions do exist). This is different from suggesting that these are two interpretations that have existed at the same time. The former point of view was increasingly unable to explain observations that were better explained with the latter point of view. And some papers made particularly important contributions to this revised interpretation, such as Battich et al., 2015, but there are many others (see also below). I suggest that the authors focus here on a particularly attractive aspect of their work, namely the multiplexed measurements with MERFISH, and state that using an image-based mRNA multiplexing approach that can also quantify cell state underscores the latter interpretation (that cell state can explain observed over-dispersion), and offers a novel generalizable approach to assess this.

On page 5, the authors state: 'As was shown in the past, cell volume strongly correlated with the total number of transcripts per cell (Figure 2D) indicating that at least for some genes cell state factors must be important contributors to their expression variability (Hansen, Desai, et al., 2018; Padovan-Merhar et al., 2015).' The authors here omit the one study that did the most extensive quantitative analysis of cell state factors in determining transcript variability across >900 human genes, by employing quantitative approaches and generating data of a quality and size that exceed the cited papers significantly (Battich et al., 2015). This cannot be omitted. In fact, the 2018 paper that is cited is rather irrelevant in this context as it is technically poorer and comes 3 years after the 2015 studies; It would be more appropriate to instead cite earlier work showing that extrinsic sources such as cell volume dominate gene expression variability (e.g. in yeast Raser and O'Shea, 2004, and Newman et al., 2006), and that in mammalian cells it has been linked to mitochondrial abundance (das Neves et al., 2010), paracrine signaling (Shalek et al., 2014), and cell cycle (Buettner et al., 2015).

On page 5 the authors also state 'Similarly, the cell cycle stage and MCF10A differentiation status were correlated with specific genes (Figure 2EF)'. This additional importance of cell cycle for explaining gene expression variability was also revealed by Battich et al. 2015, and also by scRNAseq, see Buettner et al., 2015. It would be good to mention this.

The results obtained with multilinear regression (MLR) modeling in Fig. 4D are very similar to results obtained by Battich et al., 2015, namely that the amount of unexplained variability approaches the Poisson limit. However, the approach used is somewhat different and sets of genes analyzed are also likely different. Clearly, this study represents an advance over the previous one in that it multiplexed 150 gene transcript measurements with MERFISH (Battich et al., 2015 did >900 genes in parallel), and their idea to include the first 2 PCs of the 150-dimensional transcript variability space to explain variability for each gene transcript separately (by including it in the MLR model) is a nice approach that is enabled by having multiplexed measurements and something that can be generalized. This should be emphasized as the main novelty in this paper. What would in addition be very useful, and which has the potential to even better explain the similarity in performance of the MLRs in this study compared to Battich et al., 2015, is if the authors can elaborate on the 'eigengenes' (the first 2 PCs) of their approach. What are the loadings? Are these dominated by genes that report on mitochondrial abundance, position in the cell cycle, cell shape or metabolic state? Mapping cell state space onto a 'small' set of transcripts that can capture the cellular state comprehensively would be a great resource for the whole single-cell biology field, especially also for the scRNAseq field, and this study can make a start of it. This is also pointed out in Moon et al., 2018, (about the potentially relatively small number of latent dimensions in the cell state manifold) and mentioned in their discussion. More extensive analysis of the data presented here and comparing it to phenotypic properties of this latent space, can allow the authors to be more concrete here.

On page 8, the authors refer to a recent paper that, through a misleading analysis, caused

considerable confusion in the field. The authors write: 'However, some of the proposed mechanisms such as nuclear export of RNA were shown to act as amplifiers of observed dispersion (Hansen, Desai, et al., 2018).' The suggestion made in this paper that noise is amplified at fast nuclear export rates, stems from normalizing simulated noise in the cytoplasm with simulated noise in the nucleus. At export rates faster than the average transcription rate, the apparent noise in the nucleus becomes attenuated, but this doesn't mean transcriptional noise becomes amplified in the cytoplasm. The appropriate normalization is to the noise produced by transcription itself. In such a comparison, the conclusion is that fast nuclear export rates do not buffer transcriptional noise (but also do not amplify it), which is in line with numerous theoretical and experimental studies that have addressed this question in the past (Xiong et al., 2009; Singh and Bokes, 2012; Battich et al., 2015; Bahar Halpern et al., 2015; Sturrock et al., 2017). I encourage the authors to do this simple simulation themselves (which they undoubtedly can do quickly) and draw their own conclusion. This has important consequences for the tone of the discussion, which I feel should be somewhat altered, as also pointed out below.

One page 8, the authors next state that: 'Therefore, the degree by which post-transcriptional mechanism can be used to reduced expression noise is an important open question. Until additional data will help clarify the ubiquity of such mechanisms, the most parsimonious interpretation is simply that RNA synthesis does not happen in large allele-specific bursts.' While I fully agree that additional data are needed and that we are only at the beginning of understanding what dampens intrinsic noise in biological systems (including gene expression), I would be hesitant to endorse this statement as such, as there is ample evidence for both transcriptional bursting and noise buffering. I would rather say that there are multiple factors at play that cause extrinsic (and thus in principle predictable) sources of variation to dominate gene expression variability. One is that bursting may not occur for all genes due to e.g. slow chromatin dynamics as observed in plants (Berry, Dean, Howard. *Cell Systems* 2017), or that bursting may not reflect intrinsic noise, but rather non-linear manifestations of extrinsic sources of regulation, for instance indicated by the fact that enhancer sequences can control burst size and frequency in a deterministic manner, which can cause correlated bursts between genes that have similar enhancers (e.g. Fukaya, Lim, Levine. *Cell* 2016). The other is that bursts can be attenuated by numerous posttranscriptional mechanisms. This includes the regulated balance between mRNA production, nuclear export, and cytosolic degradation rates (see for instance a very recent paper by Xiaowei Zhuang's group in PNAS, Xia et al. 2019, employing 10,000plex MERFISH revealing that the transcripts of 15% of protein-coding genes are retained in the nucleus of mammalian tissue culture cells; or a recent paper by Bas van Steensel's group in PLoS genetics, Chen and van Steensel, 2017, showing that for 90% of genes in *D. melanogaster* cells, mRNA nuclear export rate is slower than turnover rate), and may also involve numerous membraneless organelles that sub-compartmentalize the nucleus and cytoplasm of mammalian cells, and which can have various noise buffering mechanisms in gene expression (and metabolism, signaling, etc.) that are still poorly explored (e.g. F. Oltsch, C. Zechner et al., *bioRxiv* 2018). These studies come on top of strategies from control theory to buffer noise, and it would be more appropriate if the discussion includes this. In the end, it only underscores the authors' findings, namely that unexplainable variability is often small.

Reviewer #2:

In the present manuscript, Foreman et al. study noise in gene expression using co-variance analysis of mRNA counts for 150 genes that are activated by a Calcium response, using highly multiplexed RNA FISH. They show that most of the variance in gene expression can be attributed to extrinsic factors such as cell cycle stage, calcium response pattern of individual cells, cell size, etc. In their system, intrinsic noise explains only a very small fraction of gene expression variability.

Major issues

1. Generality of the main conclusions.

i) The authors claim that most variability in mRNA numbers is explained by deterministic factors, based on their results. However, the generality of this conclusion is very doubtful. The authors measure expression levels/variability of genes after their transcriptional induction by an external signal. Therefore, this is a particular out-of-steady-state situation that is not comparable to

fluctuations in transcriptional activity as they have generally been described as transcriptional bursting by most studies in the field (Raj et al., PLoS Biology 2006, Chubb et al., Current Biology 2006, Suter et al., Science 2011, etc.). In fact, Molina et al. PNAS 2013 have shown that external stimuli can strongly constrain variability in transcriptional activity between cells. Therefore, what Foreman et al. most likely observe here is a near-maximal activation of transcription of their target genes, resulting in a strong constraint on transcriptional activation. Thus, the general statement in the title "Mammalian gene expression variability is explained by underlying cell state" is not supported by this study.

ii) The claim made by the author that "the most parsimonious interpretation is simply that RNA synthesis does not happen in large allele-specific bursts" is disparaging a large number of previous work showing direct evidence of transcriptional bursting. The authors do not offer a convincing explanation why all this body of work got it wrong. Importantly, it was shown that upon stimulation by external signals, virtually all cells can respond simultaneously in their transcriptional activity (Molina et al., PNAS 2013), thus exhibiting synchronised bursting activity, in contrast to steady-state conditions in which bursts are not synchronised between cells. In their system, they observe a single strong period of gene activity after gene activation, thus providing only very limited general insights into transcriptional bursting.

2. Originality of the findings

Battich et al., Cell 2015 have shown results along the same line and explained a large fraction of intercellular variability in mRNA counts by extrinsic noise. However, despite using > 100 of different cellular parameters (Foreman et al. only used 13) they were able to explain a much smaller fraction of gene expression variability. This could be due to their study of gene activity in steady-state conditions, in contrast to the present study. It is therefore unclear how the study by Foreman et al. advances our general understanding of transcriptional noise as compared to Battich et al. and previous studies in the field.

Other issues

1. The genes that were analysed are not listed anywhere in the paper
2. In Fig.4D, the log/log scale is misleading, in fact the deviation from the Poisson limit is substantial for a number of genes (a factor of 2 - compatible with transcriptional bursting).
3. This study uses a single experimental method to draw their conclusions; live imaging of transcription using for example MS2 reporters integrated in a couple of genes would help make their quantifications of bursting more convincing

Reviewer #3:

In this manuscript, Foreman and Wollman combine deep cellular phenotyping, via live-cell imaging, with highly multiplexed single-cell gene expression measurements, via multiplexed error robust fluorescence in situ hybridization (MERFISH), with the goal of understanding what fraction of gene expression variation can be explained by a hidden cell state and what fraction can be explained by non-poisson noise in gene expression. The authors measure a series of cellular properties, including multiple aspects of Ca²⁺ transients, in a cell culture line using live-cell fluorescent microscopy. They then fix these cells, and characterize the expression of 150 different genes, simultaneously within the same cells using MERFISH. Using these measurements, they then use regression to determine what fraction of the variation observed in cell-to-cell gene expression can be explained by each of the different cellular properties they measured. They find that the vast majority of gene expression variation can be explained by these measures of cell state and that once this variation is removed the residual variation observed for each gene approaches that expected from a Poisson distribution. They conclude that, at least for the Calcium-associated genes that they measure with MERFISH, the majority of the variation in gene expression arises from hidden cell states and, thus, care should be taken when using such data to support models of 'bursty' transcription.

This manuscript is topical and of widespread interest for several reasons. First, MERFISH is

emerging as an exciting tool for highly quantitative, multiplexed gene expression measurements. By demonstrating that this technique can be combined with live-cell imaging, in this case of a fluorescent Ca²⁺ readout, the authors have substantially extended the power of MERFISH. It should also be noted that, to the best of my knowledge, the authors are the first laboratory to establish MERFISH outside of the laboratory in which it was invented. Demonstrating that this technique can be recapitulated by others is another important technical contribution of this manuscript. Second, the authors have tackled an important and long-debated topic: the origins of noise in gene expression. And while the authors are careful to acknowledge that their results may not apply to other classes of genes, their finding that much of the variation in gene expression can be attributed to hidden cellular states rather than bursty transcription certainly raises important concerns of which those studying noise in gene expression should be aware.

I support publication of this manuscript.

However, I have a few minor concerns that the authors may wish to address during their revision and resubmission:

One important concern with noise in gene expression measurements is the 'detection efficiency' of the technique. Imagine if one had a technique that only detected 10% of the molecules in the cell. If the probability of detecting any given molecule was constant, the final measured gene expression noise would be the convolution of the true biological variability and the Poisson process associated with this detection event. MERFISH has been reported to have a near 100% detection efficiency; thus, it is likely that this point is not a substantial concern. However, could the authors comment on their detection efficiency? For example, the error-correction abilities of MERFISH allow one to estimate what fraction of RNAs would not be properly detected from the per-bit error rates. Presumably, these would be easy values for the authors to calculate and report.

Similarly, the Zhuang laboratory reports a variety of metrics for the performance of MERFISH, including the per-bit error rate, the average brightness of RNA spots, the average size of RNA spots, and the thresholds applied to distinguish real spots from background. In addition, the Zhuang laboratory typically leaves a portion of the possible barcodes unused. Thus, detection of these barcodes serves as a direct measure of false positive rates. It would be very useful if the authors could provide these metrics for their own data.

1st Revision - authors' response

1st November 2019

Below we detail the revision made in response to reviewer comments:

This is an exciting paper that employs multiplexed single-molecule FISH to study sources of variability in mRNA abundance. I find the work of very high quality, and the results interpreted justly and of great significance to the quantitative single-cell biology community, and thus highly appropriate for a journal such as *Molecular Systems Biology*.

We thank the reviewer for his/her support!

On page 3, the authors state that: 'Overall both interpretations, bursting and cell state, can explain the observed over-dispersion and it is currently unclear which one is correct.' I understand why the authors pose the problem like this, as it is a nice lead into their ambition to provide insight into which interpretation is correct. But in essence, the former point of view has been systematically revised by many studies into one that more fits the latter point of view (exceptions do exist).

We revised the text to reflect this point. Specifically, we state that "There is mounting evidence, that for at least many genes most of the over-dispersion is explained by cell state variables rather than intrinsically noisy

transcriptional bursting (Battich et al. 2015). Nonetheless, the transcriptional bursting model is still widely used (Larsson et al. 2019; Ochiai et al. 2019) calling for more systematic investigation.”

On page 5, the authors state: 'As was shown in the past, cell volume strongly correlated with the total number of transcripts per cell (Figure 2D) indicating that at least for some genes cell state factors must be important contributors to their expression variability (Hansen, Desai, et al., 2018; Padovan-Merhar et al., 2015).' The authors here omit the one study that did the most extensive quantitative analysis of cell state factors in determining transcript variability across >900 human genes, by employing quantitative approaches and generating data of a quality and size that exceed the cited papers significantly (Battich et al., 2015). This cannot be omitted.

Citation updated.

On page 5 the authors also state 'Similarly, the cell cycle stage and MCF10A differentiation status were correlated with specific genes (Figure 2EF)'. This additional importance of cell cycle for explaining gene expression variability was also revealed by Battich et al. 2015, and also by scRNAseq, see Buettner et al., 2015. It would be good to mention this.

We now include a sentence saying that:
“These results are consistent with previous work demonstrating widespread cell cycle and differentiation related variability in the transcriptome (Battich et al. 2015).”

Mapping cell state space onto a 'small' set of transcripts that can capture the cellular state comprehensively would be a great resource for the whole single-cell biology field, especially also for the scRNAseq field, and this study can make a start of it. This is also pointed out in Moon et al., 2018, (about the potentially relatively small number of latent dimensions in the cell state manifold) and mentioned in their discussion. More extensive analysis of the data presented here and comparing it to phenotypic properties of this latent space, can allow the authors to be more concrete here.

We discussed this point in depth and have decided not to include additional analysis related to “effective dimensionality” of cell state space. This is a very important problem that we completely agree should be addressed and that our data can shed some light on it. However, to do full and rigorous analysis of this point will distract the key message of this manuscript. We are already working on additional paper that will address some of these points and we would prefer to keep a clean separation between these two papers.

On page 8, the authors refer to a recent paper that, through a misleading analysis, caused considerable confusion in the field. The authors write: 'However, some of the proposed mechanisms such as nuclear export of RNA were shown to act as amplifiers of observed dispersion (Hansen, Desai, et al., 2018).'

We are well aware of the disagreement between Hanswen et al 2018 and Battich et al., 2015. In an attempt to avoid the perception that we are “taking sides” we focus our analysis and discussion on our own data. We do point out in the discussion that our work is more aligned with Battich et al. However, we prefer to avoid calling previous work “misleading” and we let the readers will judge for themselves how these papers should be considered given our results.

One page 8, the authors next state that: 'Therefore, the degree by which post-transcriptional mechanism can be used to reduced expression noise is an important open question. Until additional data will help clarify the ubiquity of such mechanisms, the most parsimonious interpretation is simply that RNA synthesis does not happen in large allele-specific bursts.' While I fully agree that additional data are needed and that we are only at the beginning of understanding what dampens intrinsic noise in biological systems (including gene expression), I would be hesitant to endorse this statement as such, as there is ample evidence for both transcriptional bursting and noise buffering.

We now revised the discussion and added these sentences to the relevant paragraph: “For different genes, there can be different effects explaining why observed cytoplasmic transcript counts are distributed approximately Poisson for most genes, despite widespread observation of bursts during transcription. Expression variability could be buffered by processes such as nuclear export (Stoeger et al. 2016; Xia et al. 2019; Chen and van Steensel 2017), bursting may not occur for all genes (Berry et al. 2017), and bursting may be linked to extrinsic fluctuations in enhancer activity rather than intrinsic noise(Fukaya et al. 2016).”

Reviewer #2:

However, the generality of this conclusion is very doubtful. The authors measure expression levels/variability of genes after their transcriptional induction by an external signal. Therefore, this is a particular out-of-steady-state situation that is not comparable to fluctuations in transcriptional activity as they have generally been described as transcriptional bursting by most studies in the field (Raj et al., PLoS Biology 2006, Chubb et al., Current Biology 2006, Suter et al., Science 2011, etc.).

The reviewer statement that the gene expression is out-of-steady-state is incorrect. To prevent similar confusion by other readers we now state that:

“Ca²⁺ signaling is fast and we can measure the overall emergent phenotype of the network in less than 15 minutes (Figure 2A), a timescale faster than that of gene expression in mammalian cells (Shamir et al. 2016)”

ii) The claim made by the author that "the most parsimonious interpretation is simply that RNA synthesis does not happen in large allele-specific bursts" is disparaging a large number of previous work showing direct evidence of transcriptional bursting. The authors do not offer a convincing explanation why all this body of work got it wrong.

We apologize to the reviewer if s/he found our results “disparaging”. It was not our intent to insult anyone. We are very aware of the disagreement between our results and the literature that attempts to measure transcriptional bursting using RNA tagging approaches such as MS2 and have dedicated a whole paragraph in the discussion to this point.

Battich et al., Cell 2015 have shown results along the same line and explained a large fraction of intercellular variability in mRNA counts by extrinsic noise. However, despite using > 100 of different cellular parameters (Foreman et al. only used 13) they were able to explain a much smaller fraction of gene expression variability. This could be due to their study of gene activity in steady-state conditions, in contrast to the present study.

Again, we refer the reviewer to our point related to steady state measurements.

1. The genes that were analysed are not listed anywhere in the paper

A new table was added to the Appendix with full information on all genes names, symbols, expression mean and variance, and result of linear regression

2. In Fig.4D, the log/log scale is misleading, in fact the deviation from the Poisson limit is substantial for a number of genes (a factor of 2 - compatible with transcriptional bursting).

We refer the reviewer to panel 4D that does not have a log-log scale and directly shows Fano factor needed to evaluate the distance from Poisson limit.

3. This study uses a single experimental method to draw their conclusions; live imaging of transcription using for example MS2 reporters integrated in a couple of genes would help make their quantifications of bursting more convincing.

Yes, there is some disagreement between some of the MS2 based estimates of transcriptional bursting and our results (although not all, see Rodriguez et al. 2019). It is not realistic to require us to redo all experiments using all approaches. It is practically impossible to tag ~150 genes with MS2 at endogenous loci.

Reviewer #3:

However, I have a few minor concerns that the authors may wish to address during their revision and resubmission:

One important concern with noise in gene expression measurements is the 'detection efficiency' of the technique.

We now include estimates for our false positive and false negative rates. We estimate our detection efficiency to be ~95% and false positive rate <1% per gene per cell. Details on these estimates are provided in the Material and Methods section.

Thank you for sending us your revised manuscript. We are now satisfied with the modifications made and we think that the study is suitable for publication.

Before we formally accept the study for publication, we would ask you to address some remaining editorial issues listed below.

Corresponding Author Name: Roy Wollman

Manuscript Number: MSB-19-9146